# COVID-19 and Multiple Sclerosis: A Complex Relationship Possibly Aggravated by Low Vitamin D Levels

**DOI:** 10.3390/cells12050684

**Published:** 2023-02-21

**Authors:** William Danilo Fernandes de Souza, Denise Morais da Fonseca, Alexandrina Sartori

**Affiliations:** 1Department of Chemical and Biological Sciences, Institute of Biosciences, São Paulo State University (UNESP), Botucatu 18618-689, Brazil; 2Laboratory of Mucosal Immunology, Department of Immunology, Institute of Biomedical Sciences, University of Sao Paulo, São Paulo 05508-000, Brazil

**Keywords:** SARS-CoV-2, COVID-19, multiple sclerosis, immunopathogenesis, vitamin D

## Abstract

Severe acute respiratory syndrome coronavirus 2 (SARS-CoV-2) is an exceptionally transmissible and pathogenic coronavirus that appeared at the end of 2019 and triggered a pandemic of acute respiratory disease, known as coronavirus disease 2019 (COVID-19). COVID-19 can evolve into a severe disease associated with immediate and delayed sequelae in different organs, including the central nervous system (CNS). A topic that deserves attention in this context is the complex relationship between SARS-CoV-2 infection and multiple sclerosis (MS). Here, we initially described the clinical and immunopathogenic characteristics of these two illnesses, accentuating the fact that COVID-19 can, in defined patients, reach the CNS, the target tissue of the MS autoimmune process. The well-known contribution of viral agents such as the Epstein-Barr virus and the postulated participation of SARS-CoV-2 as a risk factor for the triggering or worsening of MS are then described. We emphasize the contribution of vitamin D in this scenario, considering its relevance in the susceptibility, severity and control of both pathologies. Finally, we discuss the experimental animal models that could be explored to better understand the complex interplay of these two diseases, including the possible use of vitamin D as an adjunct immunomodulator to treat them.

## 1. Introduction

COVID-19 is a disease caused by the SARS-CoV-2 virus infection whose severity varies depending primarily on host conditions and specific mutations found in the several virus variants. Even though the lungs are primarily affected, other organs and tissues such as kidneys, heart, and the nervous system can be injured. Most of its pathophysiological findings have been attributed to a hyperinflammatory syndrome resulting mainly from a dysregulated innate immune response. This review focuses on the interrelationship between COVID-19 and multiple sclerosis (MS), an autoimmune pathology that damages the central nervous system (CNS). Initially, we will discuss the main COVID-19 clinical manifestations, its etiologic agent and immunopathogenesis, and the nervous system involvement which takes place in some patients. MS clinical manifestations, the most relevant stages of its immunopathogenesis and the recognized relevance of viral infections to its development will then be described. The possible role of low vitamin D levels for the development and severity of both diseases and the recommendation of the patient’s supplementation to improve its anti-inflammatory potential will also be addressed. The final section of this review will be dedicated to briefly describe the current animal disease models that could be employed to investigate, simultaneously, both diseases or some aspects related to their immunopathogenesis.

## 2. COVID-19: Clinical Manifestations, Etiology, and Immunopathogenesis

A novel coronavirus named 2019-nCoV or SARS-CoV-2 was originally detected in Wuhan, China, in 2019, but in a few months, it spread out to most countries, initiating a pandemic. This viral agent causes COVID-19 which predominantly affects the respiratory system, causing flu-like symptoms such as fever, cough, sore throat, dyspnea and fatigue [1]. Variable disease severity is observed among patients; around 80% of the affected patients display mild symptoms or can even be asymptomatic, while about 15% may develop more severe symptoms. The remaining 5% of patients can evolve to severe pathological conditions characterized by acute respiratory distress syndrome (ARDS), septic shock, and multiorgan failures associated with an elevated risk of death [2]. Evolution to more severe conditions has been especially linked to advanced age, existence of comorbidities such as hypertension, diabetes and heart diseases, and genetic and epigenetic factors [3,4,5]. Although this infection is mostly characterized by a significant respiratory impairment, it can also trigger several extrapulmonary manifestations including thrombotic complications, myocardial dysfunction and arrhythmia, acute coronary syndromes, kidney injury, gastrointestinal symptoms, hepatocellular lesions, hyperglycemia and ketosis, neurologic alterations, visual disturbances and dermatologic complications [6,7].

The pulmonary and extrapulmonary manifestations of COVID-19 have been mainly attributed to a direct virus damage, given that ACE2 (angiotensin-converting enzyme), which is the entry receptor for the SARS-CoV-2, is expressed in the lungs and in these other extrapulmonary tissues. Analogously to other coronaviruses, SARS-CoV-2 consists of four structural proteins: spike (S), membrane (M), envelope (E) and nucleocapsid (N). The spike protein comprises two functional subunits: S1, which binds to the target cell, and S2, which triggers the fusion between the viral and the target cell membrane. SARS-CoV-2 uses two host proteins to enter the target cell; the ACE2 that is used for the attachment to S1 and the transmembrane serine protease 2 (TMPRSS2) that activates the protease activity of S2. For detailed information about the molecular mechanisms involved in this initial interaction between the virus and the target cell, see [8]. An overview of the viral structure and the initial process of interaction with pneumocytes are illustrated in Figure 1A,B, respectively.

The first line of response against pathogens, including SARS-CoV-2, is the innate immunity. In the case of SARS-CoV-2, its recognition by tissue-resident immune cells within the lung provides a local immune response resulting in the recruitment of further cells from the blood. Innate immune cells, including monocytes, macrophages, polymorphonuclear cells (PMNs) and innate lymphoid cells (ILCs) express pattern recognition receptors (PRRs) which identify pathogen-associated molecular patterns (PAMPs) and danger-associated molecular patterns (DAMPs). SARS-CoV-2 is able to initiate the activation of innate immunity by interacting with various PRRs, especially toll-like receptors (TLRs), retinoic acid-inducible gene 1 (RIG)-like receptors (RLRs), nucleotide-binding oligomerization domain (NOD)-like receptors (NLRs) and inflammasomes. PRR signaling triggered by SARS-CoV-2 induces the concurrent release of IFNs, mainly I and III types, and other pro-inflammatory cytokines such as TNF-α, interleukin-1 (IL-1), IL-6 and IL-18 [9]. Together these cytokines will induce antiviral programs in target cells and potentiate the specific immune response, which will eventually control the infection [10]. Contrasting with this well balanced and effective innate immunity, the evolution of SARS-CoV-2 infection to a severe condition has been associated with a reduced or delayed type I IFN response together with high levels of other pro-inflammatory cytokines and high viral titers [11,12]. This defective IFN response has been attributed to inborn errors of type I IFN immunity [13], and to the presence of autoantibodies against this cytokine [14]. Interestingly, a sustained increase in the levels of type I IFN in a later phase of the infection can also promote a poor clinical outcome [15]. Indeed, the signaling mechanisms involved in early (beneficial) or delayed (deleterious) type I IFN production are distinct. A rapid detection of viral RNA by TLR3, 6, and 7 and RLRs triggers a protective response, whereas a later activation of the cGAS-STING by DNA leads to cell death and a damaging production of type I IFN.

Much of COVID-19 severity has been attributed to immune dysregulation manifested by a low production of interferons, remarkable inflammatory response and delayed adaptive immune response. This subject has been intensively investigated and reviewed [16] and will be mentioned here only briefly to reinforce possible connection routes between COVID-19 and MS. The hallmark of most severe cases of COVID-19 is a strong inflammatory process that may ultimately lead to organ failure and patient death (as summarized in Figure 2).

The cytokine storm, also known as cytokine release syndrome is characterized by the extensive activation of macrophages, dendritic cells (DCs), NK, B and T cells and the subsequent production of high levels of TNF-α, IL-1β, IL-6, IL-12, IL-18, IL-33, IFN-I, IFN-γ, CCL2, CCL3, CCL5, CXCL8, CXCL9 and CXCL10 [17]. Several components are probably involved in the cytokine storm associated with SARS-CoV-2, including the interaction of viral RNA and proteins with PRRs, the binding of the virus to ACE2 and inflammasome activation [18]. It has been demonstrated that in human and mouse epithelial cells the SARS-CoV-2 spike protein binds to TLR2 and induces inflammation via the activation of the NF-κB pathway. The interaction of ACE2 with SARS-CoV-2 is followed by its reduced expression on the surface because it is internalized together with the virus. As the biological function of ACE2 is to inactivate angiotensin II, there is an increased serum level of this molecule. Increased angiotensin II contributes to COVID-19 severity by inducing specific autoantibodies whose presence correlates with blood pressure dysregulation [19] and increased cytokine production [20]. This strong inflammatory process is largely mediated by the NLRP3 system that promotes inflammation through the cleavage and activation of specialized molecules, including active caspase-1 (Casp1p20), IL-1β, and IL-18. The analysis of samples from moderate and severe COVID-19 cases indicated active NLRP3 inflammasome in PBMCs and tissues of postmortem patients [21]. These authors also observed the correlation of serum inflammasome-derived products, such as Casp1p20 and IL-18 with a poor clinical evolution. According to these authors, inflammation prompted by NLRP3 is initiated by IL-1β which induces the secretion of TNF-α, IL-6 and IL-8 by monocytes. These cytokines determine the influx of PMNs into lung tissue, gasdermin D activation and the subsequent formation of neutrophil extracellular traps (NETs), which can recruit platelets and promote hypercoagulability. This crucial inflammasome role in COVID-19 pathogenesis has been investigated as a potential target for therapy. To that end, a plethora of inflammasome inhibitors, including natural products as well as already authorized drugs, should be tested in pre-clinical and clinical studies [22]. Even though future studies are still required, clinical findings obtained in a randomized and double-blind placebo-controlled trial in which mefenamic acid was administered to ambulatory patients [23], showed that it significantly reduced their symptomatology in comparison to the placebo group. This efficacy was attributed to both the anti-viral and the anti-inflammatory properties of mefenamic acid.

Concerning the NETs mentioned above, their formation is even more accentuated in most severe COVID-19 cases, what has been attributed to increased immature PMNs and the presence of anti-NET antibodies. In addition, these antibodies can impair NET clearance and possibly enhance virus-mediated thrombo-inflammation [24]. IL-1β and IL-6 can also directly contribute to coagulation in the lung vasculature by decreasing adherens junctions in endothelial cells. Tissue factor positive extracellular vesicles (EVs) released by pyroptotic monocytes can also directly activate the clotting cascade and promote coagulation in COVID-19.

In addition to the direct virus damage and the deleterious immune response, pieces of evidences reinforce the view that the gut–lung axis will affect both the susceptibility and efficacy of the immune response against the virus. It is well known that the virus affects mainly the respiratory system; however, the gastrointestinal system is also a critical target. Gastrointestinal manifestations such as nausea, vomiting and diarrhea are present in a high percentage of COVID-19 patients. These symptoms have been attributed to the infection of gut epithelial cells by SARS-CoV-2 and the local dysbiosis characterized by alterations in microbiota bacterial composition and diversity [25]. The respiratory tract has its own microbiota and it was already demonstrated that infections by other respiratory viruses induce local inflammation which contributes to gut dysbiosis [26]. A similar effect could be expected from a lung SARS-CoV-2 infection.

Many patients have reported the persistence of symptoms as fatigue, exercise intolerance, dyspnea, muscle pain, insomnia, chest pain, anosmia, cough, and brain fog after the acute disease stage [27]. This condition has been denominated Post-COVID-19 syndrome or Long-COVID-19. Interestingly, in addition to the degree of infection severity, antibiotic usage has been considered one of the main risk factors for Long-COVID-19 development [28]. Antibiotic prescription, which is expected to be more common in severe COVID-19 patients, would alter the gut microbiota composition. This hypothesis is strongly sustained by evidence showing that antibiotics are major disruptors of gut microbiota [29]. In addition, gut dysbiosis triggered by excessive antibiotic administration, together with poorly controlled glycaemia and not well-regulated steroid administration were also identified as risk factors for COVID-19-associated mucormycosis [30], a rare and lethal fungal infection. Despite the complex gut dysbiosis scenario induced by the virus itself, as demonstrated in both a hamster experimental model and human patients [31,32], which is aggravated by antibiotic use, there is already a variety of promising microbiota-oriented strategies being suggested as prophylactic or therapeutic interventions such as probiotics, prebiotics, microbiota-derived metabolites and even fecal transplantation [33]. Notably, whether lung dysbiosis associated with SARS-CoV-2 infection or antibiotic usage impacts the poor outcomes of COVID-19 is still an open question. Besides the microbiota-mediated gut–lung communication axis, another important systemic axis of immune communication impacts COVID-19 and MS: the gut–brain axis, as addressed afterward in this review.

## 3. Neurological Involvement Associated with COVID-19

The neuropathology associated with COVID-19 is a complex condition related to the local presence of the virus, the induced local and peripheral immune responses and to alterations in the microbiota, the vascular and the coagulation systems. The following neurological manifestations have been reported in COVID-19 patients: headache, myalgia, dizziness and fatigue, described as mild; hyposmia, hypogeusia, visual disturbances, encephalopathy, epilepsy, paralysis and consciousness disorder, identified in moderate to severe cases; and cerebrovascular events, acute necrotizing encephalopathy, meningitis, encephalitis and Guillain-Barré syndrome, considered as severe conditions [34]. The prevalence of these manifestations seems to be particularly increased in hospitalized patients [35]. The pathogenesis of CNS infection by SARS-CoV-2 and the neurological complications are still poorly understood. Most of these symptoms have been attributed to the ingress of the virus into the nervous system. Neuro-invasion by SARS-CoV-2 has been confirmed by the virus detection in the cerebrospinal fluid of a patient suffering from Guillain-Barré syndrome [36], in infected brain organoids, in mice expressing human ACE2 and autopsies from deceased patients [37]. Two major routes have been associated with this neuro-invasion: through peripheral neurons and by hematogenous dissemination [38]. The peripheral nerve endings are believed to be the most common route used by SARS-CoV-2 to reach the CNS. The olfactory nerve is considered the major candidate because it is located very close to the olfactory epithelium which, by expressing ACE2 and TMPRSS2, allows initial virus replication [39]. Indeed, in a non-human primate model, it was demonstrated that SARS-CoV-2 can invade the CNS primarily via the olfactory bulb [40]. This process has been described as transcribrial route because it occurs across the cribriform plate of the ethmoid bone, followed by retrograde viral spread via transsynaptic transfer using an endocytosis or exocytosis mechanism and a rapid axonal transport. The virus could also gain access to the CNS via the vagal afferents from the upper airways [41] and the enteric nervous system [42]. Another possible route is through the virus’s presence in the bloodstream from where it can reach the nervous system by a direct interaction with brain capillary cells or by means of an infected leukocyte. The detection of viral particles in capillary endothelial cells in the front lobe tissue obtained in a post mortem sample gives support to this interaction with endothelial cells [43]. Infected leukocytes could also pass through the blood–brain barrier (BBB), acting as a Trojan horse. More recently, this hypothesis that infected cells could cross the BBB as a Trojan horse has been extended to include exosomes and high-density lipoproteins associated with SARS-CoV-2 [44].

The most recurrent neuropathological findings in COVID-19 patients include microglial activation, lymphoid inflammation with a clear predominance of TCD8+ cells, astrogliosis, myelin loss, hypoxia-ischemic changes, brain infarcts and hemorrhage and mi-crothrombi [45]. Part of these findings is due to the SARS-CoV-2 infection of microglia and neurons which express different receptors for spike as ACE2, ephrin (Eph) ligands and the Eph receptors, neuropilin 1 (NRP-1), P2X7 and CD147 [46]. Similar to peripheral infection, CNS infection also triggers a cytokine and chemokine avalanche causing neurotoxicity, disruption of the neuroglia homeostasis and neuronal death [47].

As previously addressed in item 2 of this review, the dissemination of the virus to the gastrointestinal system is an aggravating condition that can also affect the nervous system due to an altered microbiota gut–brain axis. It is of note that the gut-microbiota signatures shared by COVID-19 patients and neurological and psychiatric disorders have been described [48]. Such signatures are characterized by a reduction in microbial diversity and richness, an expansion of opportunistic proinflammatory pathogens and a reduction in anti-inflammatory-promoting bacteria. One of the consequences of this disrupted axis is a decreased secretion of short-chain fatty acid (SCFA), whose anti-inflammatory ability is well recognized. Therefore, the potential benefit of direct SCFA supplementation or reliance on probiotics prescription is being suggested for COVID-19 patients [49]. The disturbed synthesis of other gut–brain axis mediators, such as cytokines, 5-hydroxytryptamine and cholecystokinin, can additionally contribute to neurological manifestations during COVID-19 [50]. Notably, the gut–brain axis is also dysfunctional in MS [51], disclosing another link in this already puzzling interplay.

## 4. Multiple Sclerosis: Clinical Manifestations and Immunopathogenesis

Multiple sclerosis (MS) is classically described as an inflammatory and demyelinating disease originating from an autoimmune disturbance. It is characterized by multifocal and scattered lesions through the grey and white matter from the brain and spinal cord. A damaged myelin sheath commonly results in vision and coordination loss, muscle weakness, stiffness and spasms, pain, and changes in bladder and bowel function. An MS diagnosis is usually based on clinical presentation, supported by neuroimaging, and in some cases, by cerebrospinal fluid analysis to search for inflammatory markers and oligoclonal antibody bands [52]. Even though MS is considered a single disease, it can manifest under different phenotypes. According to [53], this characteristic is due to its multifactorial etiology that includes a genetic predisposition together with environmental elements such as infectious agents, mainly viruses, and vitamin D (vitD) levels, as will be commented on later. Understanding how these factors affect this disease is fundamental because this is a handicapping pathology whose incidence and prevalence are increasing worldwide. Four basic disease courses, which can also be referred to as phenotypes or types, are recognized: clinically isolated syndrome, relapsing-remitting MS (RRMS), secondary progressive MS (SPMS) and primary progressive MS (PPMS). RRMS is considered the most common phenotype, affecting around 85% of patients. As indicated by its designation, it is characterized by alternating relapses and remissions, which are periods of neurological dysfunction or absence of neurological symptoms, respectively [54]. A noteworthy finding is the strong association of relapses with infections, postpartum period, genetic risk factors, stress, and vitD levels [55]. In addition, an increased relative risk for relapses has been associated with infections located in the upper respiratory system or affecting the urinary and gastrointestinal tracts [56].

MS immunopathogenesis has been disclosed by using both patient samples and information from animal models, especially experimental autoimmune encephalomyelitis (EAE). This disease is induced in mice and rats by immunization with myelin-derived proteins and peptides in the presence of complete Freund’s adjuvant and pertussis toxin [57]. The sequence of events leading to the onset of MS is outlined in Figure 3 and is briefly described below.

### 4.1. Peripheral Activation of Myelin-Specific Lymphocytes

It is accepted that the activation of myelin-specific T cells would occur in peripheral lymphoid organs by different mechanisms such as the recognition of microbial epitopes sharing homology with self-antigens (molecular mimicry), the release of myelin from the CNS by a local insult, or by bystander activation [58]. The perspective that molecules derived from a dysbiotic gut microbiota could induce neuroinflammation and symptoms of MS by mimicking autoantigens has been interestingly proposed [59]. Other mechanisms, including the induction of co-stimulation, polyclonal activation, altered processing and expression of cryptic antigens could also be induced by infectious agents and contribute to the onset of autoimmunity [60].

### 4.2. Presumed T Cell Licensing

A fairly new concept has emerged from EAE studies in the last few years and is called “licensing” or more clearly “licensing for pathogenicity”. This stage, which seems to occur in the lungs, spleen or maybe both, allows the T cells to become pathogenic, reach the CNS and orchestrate a local inflammatory process [61,62]. This licensing process is accomplished by a change in the gene expression pattern marked by the downregulation of proliferation/activation-related genes and upregulation of migration-promoting genes [63]. Th17 cells play a critical encephalitogenic role by opening the BBB [64] and promoting neurodegeneration [65].

### 4.3. Expansion of Th17 in the Intestine

The immune system associated with the intestinal mucosa has recently been recognized as pivotal to MS and EAE development. This crucial role in the pathogenesis occurs mainly by promoting the activation and acquisition of the Th17 phenotype [66,67]. It is assumed that the activation of effector Th17 cells occurs mostly in the murine small intestine, regardless of their ensuing function [66,68]. Interestingly, the development of steady-state or pathogenic Th17 cells is critically determined by microbiota composition. Segmented filamentous bacteria, for example, induce brain autoimmunity in mice by selectively privileging Th17 differentiation [69].

### 4.4. Breakdown of the Blood Barriers and Cell Migration to the CNS

Two barriers protect the CNS integrity and functionality: the BBB and the blood–cerebrospinal fluid (B–CSF) barrier. They are located in distinct CNS compartments and their dysfunction can allow leukocyte access and the ensuing neurodegeneration in MS and EAE [70]. The integrity of these barriers can be disturbed by peripheral inflammation, infections and proinflammatory cytokines. There are a number of mechanisms by which an exaggerated inflammatory process can induce the disruption of these barriers, including changes in tight junctions, damage to endothelial cells, the activation of astrocytes and microglia, and the effects of peripheral immune cells [71]. The invasion of the CNS by neurotropic viruses, by hematogenous or non-hematogenous routes, can be associated with structural and functional BBB alterations that also lead to its breakdown [72]. Specific cytokines have been more often linked with alterations in these barriers as shown by [73], which found that periodontal inflammation-induced IL-6 is associated with neuroinflammation and BBB disruption in the hippocampus.

### 4.5. Local Inflammation and Neurodegeneration

A variety of infiltrating cells, mainly γδT, Th1 and Th17 cells, are locally expanded and release cytokines that will activate microglia and oligodendrocytes. Together with the ones mobilized from the periphery, these activated cells will release inflammatory mediators such as IL-8, IL-17, GM-CSF, CCL2 and enzymes that will trigger neurodegeneration [74,75]. A great deal of contribution to this damaging process has been imputed to B cells, mitochondrial dysfunction, oxidative stress and inflammasome activation [76,77]. This process will eventually be controlled by regulatory mechanisms involving different cell subsets such as Tregs CD25+Foxp3+, Tr1, Qa-1 restricted CD8, regulatory B cells, NK and CNS-derived myeloid-derived suppressor cells [78]. However, defined viruses, namely EBV and HHV6, could trigger relapses through a peripheral mechanism rather than a direct effect through intrathecal multiplication [79].

## 5. Connection between MS and Virus (EBV and SARS-CoV-2)

The Epstein-Barr virus (EBV) establishes lifelong infection, usually subclinical, in more than 90% of the adult population worldwide. However, it is also the causal agent of infectious mononucleosis, some types of cancer and severe lymphoproliferative diseases. Epidemiological, serological and virological pieces of evidences support its role also in MS development [80]. Primary EBV infection occurs in the squamous epithelial cells where it replicates and from where it reaches and infects the B lymphocytes from Waldeyer´s tonsillar ring. Consequently, naive B cells undergo a germinal center-like activation and differentiation program affecting more than 11,000 genes. This process culminates in proliferating immortalized B cells resembling plasmablasts and early plasma cells [81]. Therefore, in the context of MS, EBV is understood as a disease trigger. This possibility was tested in a large cohort of more than 10 million US army personnel [80]. A 32-fold increase in MS diagnosis in individuals who became seropositive, compared with those that remained seronegative, was observed. This role of EBV has been mainly attributed to cross-reactivity between self and EBV antigens, involving both cellular and humoral immunity [82]. Additionally, as B cells and plasma cells have been identified in the brain of deceased MS patients, but not in controls [83], it is believed that induction of B cell trafficking to the CNS is also involved in this mechanism [84]. The recognized efficacy of monoclonal antibodies such as rituximab, which targets the B cell surface marker CD20, adds more credibility to the contribution of EBV to MS development [85]. Considering that certain deregulated immunological pathways found during severe COVID-19 coincide with immune alterations present in MS, it has been postulated that SARS-CoV-2 could be a risk factor for the triggering or worsening of MS in prone individuals.

Through a system biology study, [86] found the expressive interaction of SARS-CoV-2 with genes associated with autoimmunity, especially MS. In this scenario, these authors highlighted three intersecting pathways: type-1 IFN response, Th17 axis, and inflammasome pathway, which were considered critical in this COVID-19 *vs* MS interplay and that will be briefly commented on. The hypothesis that MS is a type of IFN I deficiency syndrome was initially proposed in [87] and later expanded to encompass other autoimmune diseases as well [87]. Type I IFN appears to play a pivotal role in the CNS, avoiding both severe infections, especially viral ones through its antiviral effect, and local inflammation through its immunomodulatory potential [88]. In 1993, type I IFN, mainly the β-1a type, was adopted as the first disease-modifying therapy for MS [89]. Coincidentally, dysregulated and/or delayed type I IFN responses are also associated with severe COVID-19 prognosis. Dysfunctional IFN I production can be caused by inborn errors, by the presence of anti-IFN I autoantibodies and by inhibition of type I IFN production by several SARS-CoV-2 proteins [90,91]. In this context, one can speculate that MS patients with IFN I deficiencies would be more susceptible to SARS-CoV-2. To the best of our knowledge, this possibility has not been investigated yet. On the other hand, MS patients under type I IFN therapy would be more protected against severe forms of COVID-19. This possibility was reinforced in [92], which showed a lower, even though not significant risk of infection by this virus in these MS patients. Moreover, only minor COVID-19 symptoms were described in an MS patient under IFN I therapy [93]. Given this scenario, we could presume that COVID-19 is potentially able, considering its ability to reach the CNS and to infect an IFN I deficient MS patient, to trigger an MS relapse or worsen disease symptoms.

The Th1/Th17 axis, which is responsible for IFN-y and IL-17 production, is also shared by both pathologies. The contribution of these Th subsets to MS immunopathogenesis is strongly supported by the literature [94]. Interestingly, the Th17 subset develops in the intestine and its higher frequency correlates with microbiota alterations [67]. Coincidentally, severe COVID-19 is associated with high levels of IFN-y, Th17 polarization [10] and gut microbiota dysbiosis [32]. This scenario suggests that a Th1/Th17 active axis in COVID-19 or MS patients could aggravate MS or COVID-19 symptoms, respectively.

Another critical link at the crossroad between COVID-19 and MS is the inflammasome, a complex molecular platform comprising a sensor protein, inflammatory caspases and, in some cases, an adapter protein that bridges the two other components. Its activation by DAMPs and PAMPs promotes IL-1β and IL-18 production and pyroptosis [95]. Dysregulated inflammasome activation can be associated with infections and inflammatory pathologies. A strong contribution has been attributed to inflammasomes, especially the NLRP3, to the development of MS and its experimental animal model (EAE). NLRP3 activation is involved in various MS stages such as initial inflammation, T cell polarization, CNS barrier breakdown and neurodegeneration [96]. The ability of SARS-CoV-2 to activate this platform has also been clearly demonstrated during COVID-19 and is accentuated in the more aggressive disease [97]. This association makes sense, considering that a significant cause of COVID-19 pathogenesis and subsequent severity is the cytokine storm associated with NLRP3 overactivation [98].

Despite the expectation of a deleterious effect of COVID-19 on MS manifestations, a retrospective study concluded that, regardless of its severity, COVID-19 was not associated with an increased risk of MS relapse shortly after infection [99]. The authors attributed these non-expected results to post-COVID-19 lymphopenia and to the use of immunomodulatory drugs to control MS. However, the authors also do not rule out the possibility that COVID-19’s deleterious effects will be observed later as sequels associated with post-COVID condition. The stages of MS immunopathogenesis that could most likely be deleteriously affected by SARS are schematically suggested in Figure 4.

## 6. Connection between Multiple Sclerosis and Vitamin D

Numerous questions have been raised concerning the relationship between MS and vitD levels. Some of the most relevant ones, considering practical purposes, are: Is there a vitD deficiency in MS patients and is this a risk factor to develop this disease and to present a more severe pathology? Is vitD supplementation indicated for MS patients? How would vitD control MS pathogenesis?

The literature data concerning the relationship between vitD and MS support the concepts that there is a vitD deficiency in these patients, that this is a risk factor for disease development and that this deficit probably contributes to a more severe pathology. The hypothesis that adequate vitD levels were relevant to preventing MS development emerged from the realization that this disease was more prevalent in geographical regions with a lower solar incidence where the production of this vitamin by the skin, stimulated by UV light, is low. The high prevalence of vitD deficiency in MS patients was formally described in 1994 [100]. This finding was subsequently validated by other authors who also showed a significant correlation between this deficiency, MRI load and disease severity [101]. Several studies have been conducted to determine if vitD supplementation could be considered as an add-on therapy for MS. A study performed in relapsing-remitting patients investigated its supplementation in patients under IFN-β-1b [102]. Even though no difference has been observed in the annual relapse rate, the vitD supplemented group presented a reduced disease activity indicated by MRI. According to [103], the supplementation with vitD of MS patients under natalizumab treatment corrected hypovitaminosis and decreased annualized relapse rate. As vitD efficacy could rely on high doses and considering that this could trigger fatigue, muscle weakness, renal failure and cardiac arrhythmia, some clinical trials were designed to test possible deleterious outcomes. It was found, however, that even vitD supplements that elevated twice the top of physiological range did not elicit hypercalcemia, hypercalciuria or any other detrimental effect [104]. Nonetheless, a clinical trial concerning the efficacy of a high vitD dose was inconclusive, neither supporting nor discrediting its potential benefit [105]. So far, there is no substantial evidence to approve this vitamin as an add-on therapy for MS [106]. Despite the literature discrepancies, these authors recommend implementing standardized study designs with well-defined variables concerning the kind of vitD supplement, its concentration, cohort composition, and the clinical and laboratory parameters to be evaluated. They also recommend the collection and storage of samples to assemble a bio-bank for further evaluations. Detailed reasoning and suggestions regarding their proposed study design are available in their publication.

Our research team has been testing vitD efficacy in EAE. Our findings indicate a clear window of opportunity for therapy with vitD in an animal model which could also be relevant for the human disease. We found that vitD is still effective when administered during the preclinical disease phase. However, it is important to clarify that earlier supplementation determines a more pronounced therapeutic effect [107,108].

It is well established that vitD signaling pathways are able to regulate both innate and adaptive immunity, keeping the associated inflammatory response within physiological limits. The immunomodulatory ability of vitD is clearly pleiotropic and reaches the majority of the immune cells in different phases of the immune response. This is explained by its interaction with the vitamin D receptor (VDR) expressed in immune cells including PMNs, macrophages, DCs, and B and T lymphocytes [109]. Understanding this essential VitD biological activity has sparked much interest in two aspects: vitD level in patients with inflammatory diseases and the possibility of its supplementation as a therapeutic measure. The interaction of the active form of vitD with VDR and the main effects over the immune system are illustrated in Figure 5. An overview of the main outcomes from vitD interaction with immune cells is presented in Table 1. Its potential as an adjunct or alternative therapy in inflammatory diseases is exemplified in Table 1.

The presumed protective effect of vitD on MS may occur in both the periphery and the CNS. As previously mentioned in this review, it has been suggested that the immune response against neural antigens is initiated in the peripheral lymphoid organs. VitD could already be effective at this initial stage by modulating the differentiation and function of APCs. It has been widely demonstrated that vitD affects the differentiation, maturation and function of DCs, directing them to a tolerogenic profile [107,142]. These APCs will then modulate naïve TCD4+ lymphocytes toward a functional hypo-reactive state. VitD-induced tolerogenic DCs are also capable of driving the differentiation of Tregs. This effect was demonstrated in EAE by the adoptive transference of vitD-induced IDO+ DCs, which are immature and tolerogenic cells. This procedure increased the percentage of CD4+CD25+Foxp3+ Tregs in the lymph nodes of rats with EAE [143]. The ability of VitD to decrease proliferation and differentiation of effector proinflammatory T cells in EAE was also demonstrated [144]. This effect was associated with the downmodulation of various metabolic and signaling routes which are essential for Th1/Th17 polarization. This inhibition was concurrent with reduced DNA methylation and the upregulation of many non-coding RNA classes.

One of the critical stages of MS immunopathogenesis is the BBB breakdown which can be detected in patients via gadolinium-enhanced MRI in the CNS [145]. An effect at this level is expected, considering that BBB endothelial cells express VDR. Our research team showed that calcitriol administration to a murine model of MS decreases neuroinflammation and reduces BBB disruption [146]. Other experimental pieces of evidence have disclosed the molecular mechanisms underlying this protective effect. By using an in vitro system comprising a human brain microvascular endothelial cell line stimulated with TNF-α or sera from MS patients, the authors in [147] showed that vitD determined the upregulation of tight junction proteins and downregulation of cell adhesion molecules. VitD can also downmodulate neuroinflammation by targeting additional CNS cells. It decreases, for example, the production of TNF-α, IL-1β and the expression of IL-4 by astrocytes stimulated with LPS in vitro. This likely outcome was reproduced in neonatal rats injected with LPS [148]. The addition of vitD to stimulated microglial cells reduces the expression of Iba-1, MHC-II, CD86 and TLR-4 in vitro, and in EAE [146,149]. By using a model of demyelination in rats, the authors in [150] described that VitD also operates at the level of remyelination. VitD promotes the proliferation and differentiation of neural stem cells and their migration to the lesion site, where they subsequently differentiate into oligodendrocyte lineage cells and produce myelin basic protein. Lastly, the possibility that vitD is also acting through inflammasome inhibition must be considered. This system operates in the periphery during the initial induction/expansion of Th1/Th17 cells and also during the inflammatory process in the CNS. The demonstration in [151] that vitD negatively regulates the NLRP3 inflammasome via VDR signaling, effectively inhibiting IL-1β secretion, gives some support to this additional level of vitD protection in MS.

## 7. Connection between COVID-19 and Vitamin D

Analogous to the approach used to analyze the relationship between MS and vitD, we also limited the complex connection between COVID-19 and vitD to answer the same three questions: Is there vitD deficiency in COVID-19 patients and is this a risk factor for getting infected and to develop a more severe pathology? Is vitD supplementation indicated for COVID-19 patients? How would vitD control COVID-19 pathogenesis?

An analysis between vitD levels and the number of COVID-19 cases in 20 European countries showed a significant negative correlation, suggesting that higher levels of this hormone could afford some protection against SARS-CoV-2 infection [152]. These findings were reinforced by the observation that many hospitalized COVID-19 patients presented vitD serum levels considered below the normal expected ones [153,154]. The majority of the findings also support an inverse correlation between vitD deficiency and a poor prognosis for COVID-19. According to [155], low vitD levels at hospital admission were associated with increased IL-6 production and predicted the severity of respiratory distress and mortality during the course of hospitalization. This association between vitD deficiency and severe COVID-19 has been confirmed by several other researchers [156,157,158,159]. From a theoretical point of view, based mainly on vitD’s immunomodulatory properties, its adoption as an adjunct therapy for COVID-19 seems consistent. For instance, the authors in [160] have investigated the effect of oral vitD supplementation in mild to moderate COVID-19 patients with low levels of this vitamin. They observed that 5000 IU of vitD reduced the recovery time related to cough and loss of taste and smell. The adequate levels of vitD in the host have been associated with the reduced release of proinflammatory cytokines, thus lowering the risk of a cytokine storm; increased levels of anti-inflammatory cytokines; and enhanced secretion of natural antimicrobial peptides. It may also be involved in the enhancement of the Th2 immune response and activation of defensive cells such as macrophages, as illustrated in Figure 5. Contrary to these findings, several studies have concluded that there is no direct association between vitD concentrations and a poor prognosis of the disease [161]. By employing a meta-analysis and GRADE assessment of cohort studies and RCTs, the authors of [162] inferred that low vitD levels do not play a role in disease severity and that supplementation does not improve outcomes in hospitalized patients.

The explanation for these conflicting results could be partially related to intra- and inter-cohort variability. Other parameters including supplementation protocols such as doses, period of supplementation, patient’s age, presence of comorbidities and even the risk of bias, could contribute to this variability. This ambiguous scenario has prevented an official recommendation concerning the prophylactic or therapeutic use of vitD for COVID-19 control [163]. The presumptive ability of vitD to control SARS-CoV-2 infectivity and COVID-19 severity would be mediated by different mechanisms. Some of them are related to the capacity of this hormone to increase the production of antimicrobial peptides, in particular cathelicidin antimicrobial peptide, also known as LL-37 [164]. LL-37 is produced by immune cells and epithelial cells from the skin and respiratory tract. Experimental data strongly suggests that this peptide can inhibit SARS-CoV-2 infection and other alterations that contribute to disease severity. Human cathelicidin can inhibit virus infection by directly interacting with the SARS-CoV-2 RDB and also by masking the ACE2 [165]. According to [166], a plethora of other biological activities have been ascribed to LL-37 and could contribute to its eventual preventive and therapeutic adoption against COVID-19. In this sense, this peptide is endowed with an immunomodulatory ability, could facilitate efficient NET clearance by macrophages and speed endothelial repair. These authors also addressed the fact that further investigations about the VitD/LL-37 axis in the context of COVID-19 are highly recommended considering that vitD could be a widely accessible strategy.

One of the hallmarks of severely affected COVID-19 patients is the presence of a cytokine storm, mainly triggered by the activation of cells from innate immunity. The well-established ability of vitD to directly control cytokine and chemokine production could provide another mechanism for vitD usefulness as an adjunct therapy for COVID-19. This effect derives mostly from the downmodulatory ability of vitD over Th1 and Th17 differentiation and cytokine production [167,168]. This mechanism is already suggested by clinical findings in COVID-19 patients. The authors in [169] described that vitD supplementation in geriatric intensive care patients suffering from COVID-19 reduced many inflammatory parameters, including IL-16, C-reactive protein, procalcitonin, D-Dimer, ferritin and lactate dehydrogenase. Its activity against endothelial dysfunction [170], and vascular thrombosis [170] could also contribute to the ability to control COVID-19 immunopathogenesis.

## 8. Experimental Animal Models to Decipher the Complex COVID-19 and MS Interplay

Many questions concerning the relationship between COVID-19 and MS have already been raised and partially answered by experts in the field. We believe, however, that experimental animal models could add more knowledge to the remaining gaps of the complex interplay between these two pathologies. Insightful and updated reviews have been published regarding the most useful animal models to investigate, separately, these two pathologies [57,171,172,173].

For the sake of objectivity, only the models that seem to be immediately or more easily available to investigate aspects concomitantly related to these two pathologies will be briefly described, as summarized in Figure 6. Syrian hamsters (*Mesocricetus auratus*) are widely used in the research of respiratory viruses. In addition, their ACE2 receptor binds tightly to SARS-CoV-2 which makes then naturally susceptible to infection by this virus. The experimental trans-nasal inoculation of SARS-CoV-2 in 4–8-week-old hamsters triggers a reproducible infection characterized by a short-term, self-limiting, epitheliotropic infection of the lungs and intestine with almost complete elimination of the virus before 14 days post infection. Details of these lesions, which are similar to the ones found in humans infected with SARS-CoV-2, were described in [174]. This experimental disease can progress with different degrees of severity, depending upon the hamster strain [175]. It was recently described in [176] that hamsters develop a condition that clearly resembles the post-acute sequels of COVID-19. After virus clearance, these animals presented a clear inflammatory process in both the olfactory bulb and the olfactory epithelium. This process included the activation of myeloid and T cells, and the production of proinflammatory cytokines, including IFN-y. We believe that this is an interesting model to be explored in the context of these two diseases. The use of hamsters to model MS are scarce. However, older publications by the authors of [177] showed that Syrian hamsters immunized with guinea pig spinal cord derived antigen, in the presence of adjuvants, developed a chronic paralysis after 50–100 days, which was often relapsing. Seventy percent of these animals presented mononuclear cell infiltration and focal demyelination in the neuraxis [178] also demonstrated that the susceptibility of these rodents to develop EAE was highly dependent on the specific inbred strain, with some being able to develop acute paralysis around 10–21 days after immunization. Interesting, this model was already used to test the possible interference of a virus on EAE development [179] showed that the persistence of the measles virus in the CNS exacerbated EAE manifestations.

For many reasons, including the availability of reagents to perform immunological characterizations, mice constitute the first choice for these investigations. However, wild-type murine ACE2 does not bind adequately to the viral spike protein, rendering them resistant to the infection [180]. Different strategies have been engendered to overcome this obstacle. We believe that transgenic mice expressing human ACE2 would be worthwhile to be tested, considering that COVID-19 severity could be controlled by the level of ACE2 expression [173] which would allow a more precise investigation about the potential role of this infection on EAE aggravation. These transgenic mice need to have a C57BL/6 and SJL background to be able to develop the classic EAE pathology. To the best of our knowledge, the suitability of ACE2 transgenic C57BL/6 and SJL mice strains to develop EAE was not tested yet. This investigation is mandatory considering that transgenesis could alter the evolution of EAE in these animals. The most employed animal model for MS studies is experimental autoimmune encephalomyelitis (EAE). Murine EAE is usually induced by active immunization with myelin-derived peptides emulsified with complete Freund’s adjuvant in the presence of pertussis toxin. C57BL/6 and SJL/J mice strains immunized with specific immunodominant peptides develop a chronic and a relapsing-remitting form of the disease, respectively [181]. EAE in mice is characterized by an ascending paralysis that starts by the tail, followed by limb and forelimb paralysis, and its clinical severity can be easily classified by a clinical score based on a five-point scale [182], together with weight loss. The immunization of SJL/J mice with PLP139-151 can result in an initial paralytic attack, followed by multiple remissions and relapses, whereas immunization of C57BL6/J mice with MOG35-55 usually causes a chronic disease course in which an initial attack does not resolve.

*Caenorhabditis elegans* (*C. elegans*) is a nematode species which has been increasingly employed as a model to investigate human diseases. This has been possible because humans and *C. elegans* share some identity concerning the digestive, the nervous and the reproductive systems. Indeed, many important signaling pathways are highly conserved between this worm and humans. Even though this worm lacks the classic adaptive immunity system, which is typical of vertebrates, it is endowed with a variety of innate mechanisms that have been studied to understand microbe-host interactions, originally during bacterial infections [183]. Later on, it was discovered that *C. elegans* could be also employed to investigate anti-viral defense mechanisms [184]. This was demonstrated by using both natural viruses such as Orsay, Santeuil and Le Blanc [185,186] and non-natural ones such as Flock House and stomatitis virus [184]. Similar to the murine models, this nematode could be adapted to SARS-CoV-2 research by expressing the human ACE2 receptor and TMPRSS2 co-factor. In the context of this review, this transgenic nematode model could be especially useful to investigate alterations in innate immunity and the nervous system associated with SARS-CoV-2 infection. As far as we know, *C. elegans* has not being directly used to study aspects related to MS; however, it is a well-established model to investigate neurodegenerative diseases in a general way, offering many advantages over other model systems to decipher the involved mechanisms. Of particular importance for the study of neurodegenerative processes are the nematode’s small genome, the anatomical simplicity and the availability of a complete 3D map of the 302-cell nervous system [184,187]. It has been suggested that the control of more severe COVID-19 cases will require a poly-therapeutic approach including both anti-viral and anti-inflammatory medicines. According to [188], *C. elegans* was recently included as an additional system to establish a combination therapy platform to treat COVID-19. Interestingly, *C. elegans* express DAF-12 that is a nuclear hormone receptor which is homologous to the VDR expressed in human cells. The authors in [189] demonstrated that the uptake of vitD by *C. elegans* via their traditional *E. coli* food source results in a significantly extended lifespan. The capacity of this worm to respond to vitD could be additionally useful in studies involving neurodegeneration considering that this hormone has a well-defined neuroprotective role [190]. The characteristics of this nematode model system which includes relative simplicity, ease of use, exquisite genetics, and an available genomic sequence, provides an extremely useful model system in many areas of study. Indeed, many important signaling pathways are highly conserved between *C. elegans* and humans; this worm has more than 7500 genes with human homologs [191].

Having in mind that most of COVID-19’s pathogenesis is due to a hyperinflammatory reaction and that this process can affect MS, a few models of inflammation induction by virus antigens are also succinctly described. Ref. [18] observed that the spike protein is able to induce inflammatory cytokines and chemokines, including IL-6, IL-1β, TNF-α, CXCL1, CXCL2 and CCL2, but not IFNs, in human cells, in mouse macrophages or lung epithelial cells. The potential of the spike protein to induce inflammation in vivo was shown by [192]. However, the most relevant data from their work was the characterization of a lung inflammation model induced by coadministration of aerosolized S protein and LPS to the lungs. This procedure triggered a strong pulmonary inflammation and a cytokine profile similar to that observed in more severe COVID-19. According to the authors, this model mimics better the more stringent lung involvement in patients with comorbidities such as diabetes, obesity and chronic obstructive pulmonary disease. These patients frequently present abnormal gut permeability allowing the translocation of LPS through the gut epithelia and, therefore, its availability to interact with the virus spike. The ability of the spike to strongly bind to LPS and boost the proinflammatory activity was previously demonstrated in [193]. The whole inactivated virus has also been employed to elicit inflammation. The intratracheal instillation of human ACE2-transgenic mice with formaldehyde-inactivated SARS-CoV-2 caused weight loss and pulmonary pathologic alterations such as consolidation, hemorrhage, necrotic debris and hyaline membrane formation. IL-1β, TNF-α, IL-6 and the infiltration of activated neutrophils, inflammatory monocytes, macrophages and T cells were also detected in the lungs [194]. We recently established a model of lung inflammation via the intranasal instillation of UV-inactivated SARS-CoV-2. This procedure triggered an exuberant inflammatory process composed of various cell types and mediators similar to lung inflammation associated with COVID-19 [195]. This inflammatory process was significantly downmodulated by intranasal vitD administration, suggesting that this hormone has the potential to be an adjunct therapy for COVID-19. In addition, considering our previous data that support a strong protective effect of vitD on EAE development [107,108,146], we believe that IN vitD administration could downmodulate inflammatory reactions occurring simultaneously in the lungs and the CNS.

## 9. Conclusions

COVID-19 and MS are associated with several immunological disturbances that could, theoretically, interfere with each other’s disease onset or outcome. SARS-CoV-2 displays, for example, molecular mimicry with CNS epitopes and causes microbiota and BBB disruption which are crucial for MS development. This virus can also reach and inflame the CNS itself which is the target of the autoimmune inflammatory reaction that characterizes MS. These two pathologies also share a possible type I IFN deficient production and hyperactivation of both the Th1/Th17 axis and the NLRP3 inflammasome platform which could mutually cause disease aggravation. The role of vitD levels in susceptibility, severity and possible adjunctive therapy in both diseases have been investigated and highly discussed but not well-established yet. This complex interplay between COVID-19 and MS urgently needs further and in-depth investigations. A plethora of experimental animal models, usually employed to study each of these pathologies individually, as is the case of *C. elegans*, hamster strains and transgenic mice, could be explored to investigate aspects related to both diseases simultaneously. These disease models could not only complement the current knowledge but also possible future questions, bearing in mind that more severe neurological changes associated with long-term COVID are possible.

## Figures and Tables

**Figure 1 cells-12-00684-f001:**
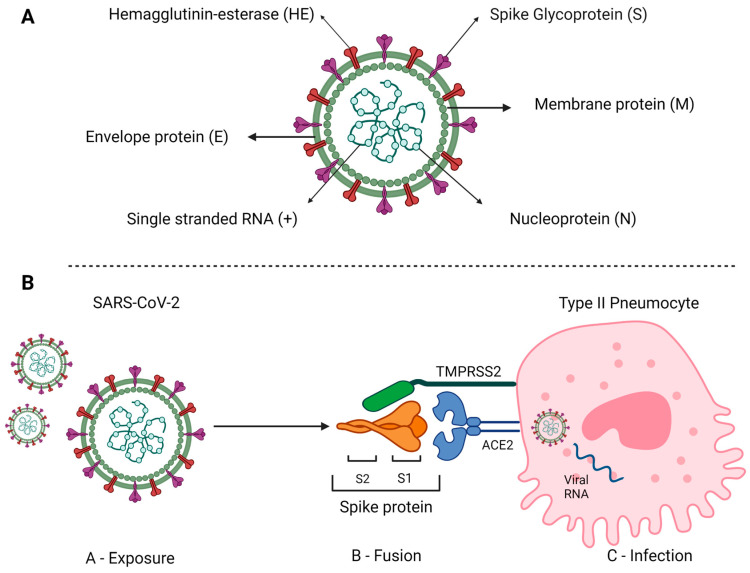
Schematic representation of SARS-CoV-2 structure and host cell invasion. (**A**) This is an enveloped, positive-sense RNA virus containing the following main structural proteins: spike (S) and membrane (M) glycoproteins, and envelope (E) and nucleocapsid (N) proteins. (**B**) Virus exposition occurs primarily through the upper airways, with tracheal and lung cells being the primary targets of infection. The interaction with these cells involves the spike protein expressed on the surface of the viral particle and comprising the S1 and S2 domains which interact with the host membrane proteins ACE2 and TRPMSS2, respectively, resulting the virus/cell fusion. Once the processes of fusion and the passage of the virus’s genetic material into the cell are completed, replication starts. Source: Created with Biorender.com.

**Figure 2 cells-12-00684-f002:**
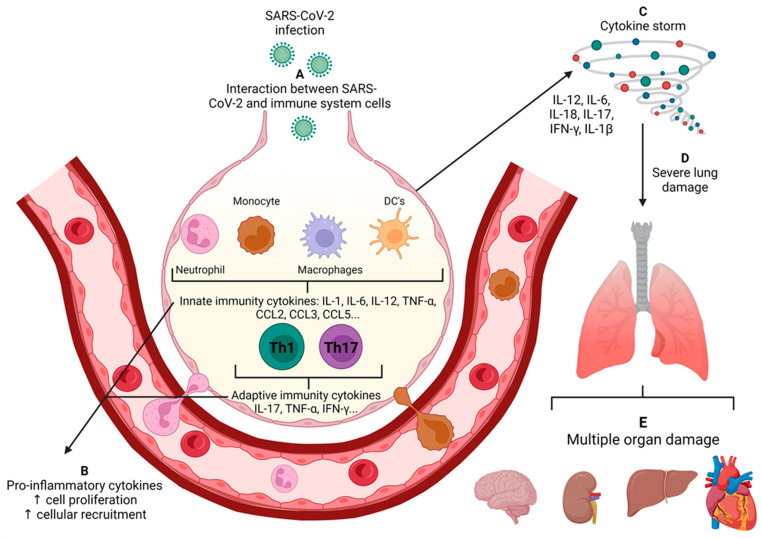
Development of systemic inflammation in the severe COVID-19. (**A**) After infecting lung epithelial cells, innate receptors distributed in the several lung immune cells recognize viral components and promote the secretion of several cytokines and chemokines. Such immune mediators act locally by promoting local inflammation and recruitment of inflammatory cells, such as neutrophils and monocytes, and by activating adaptive immunity. (**B**) Finally, in the severe forms of the disease, the overproduction of inflammatory mediators and lack of regulatory mechanisms favor the dissemination of the inflammatory process, which becomes systemic. (**C**) This process is called a cytokine storm, which drives the development of severe lung damage (**D**) and multiple organ dysfunction (**E**). Source: Created with Biorender.com.

**Figure 3 cells-12-00684-f003:**
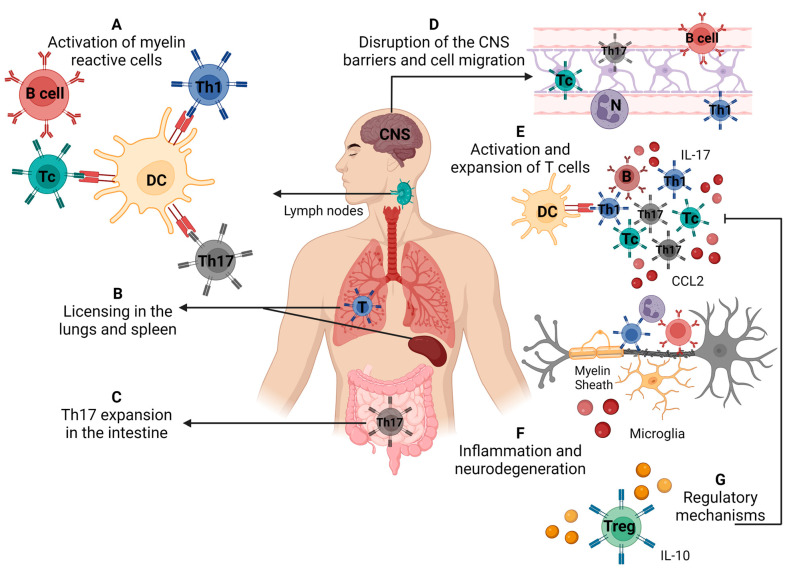
Immunopathogenesis of multiple sclerosis/experimental autoimmune encephalomyelitis. (**A**) Activation of self-reactive T cells specific for myelin antigens in secondary lymphoid organs, (**B**) licensing of self-reactive cells in the lungs and spleen (**C**), differentiation of Th17 cells in the intestine, (**D**) disruption of the blood–brain barrier and cell migration to the central nervous system, (**E**) local reactivation and expansion of Th cells, (**F**) local inflammatory process that leads to demyelination and neurodegeneration, (**G**) cells and molecules that mediate the control of disease via regulatory mechanisms. Source: Created with Biorender.com.

**Figure 4 cells-12-00684-f004:**
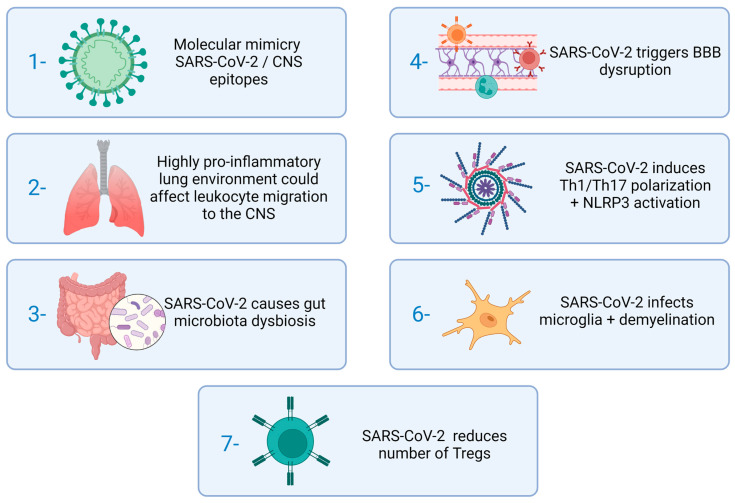
Stages of MS immunopathogenesis which could be affected by SARS-CoV-2. Source: Created with Biorender.com.

**Figure 5 cells-12-00684-f005:**
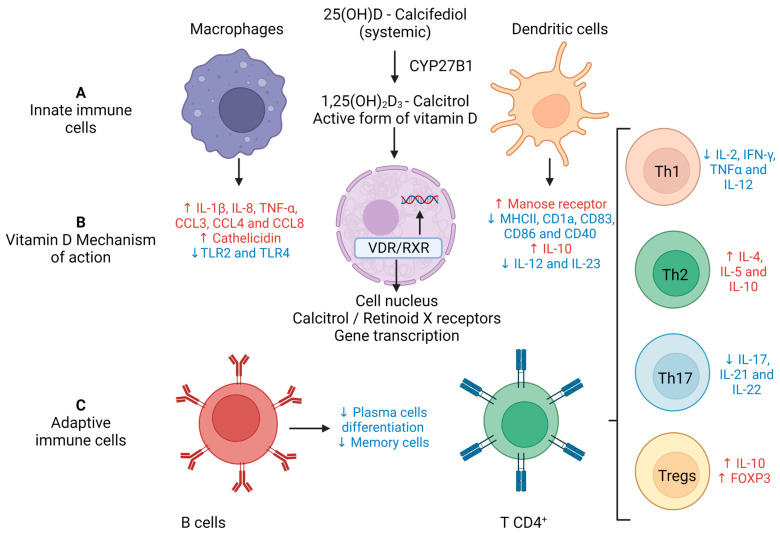
Immunomodulatory effects of vitamin D on innate and adaptive immunity. (**A**) Calcifediol (25(OH)D3) becomes biologically active forming calcitriol (1,25(OH)2D3) through two consecutive hydroxylations, with the last one being performed by 1α-hydroxylase (CYP27B1) which is present in numerous cells of innate immunity. (**B**) Calcitriol’s biological actions are mediated through binding to the VDR. This binding induces a conformational change that facilitates interaction with RXR and the coregulatory complexes required for the transcription of target genes. (**C**) Downmodulatory effects of vitD on adaptive immunity. Source: Crated with Biorender.com.

**Figure 6 cells-12-00684-f006:**
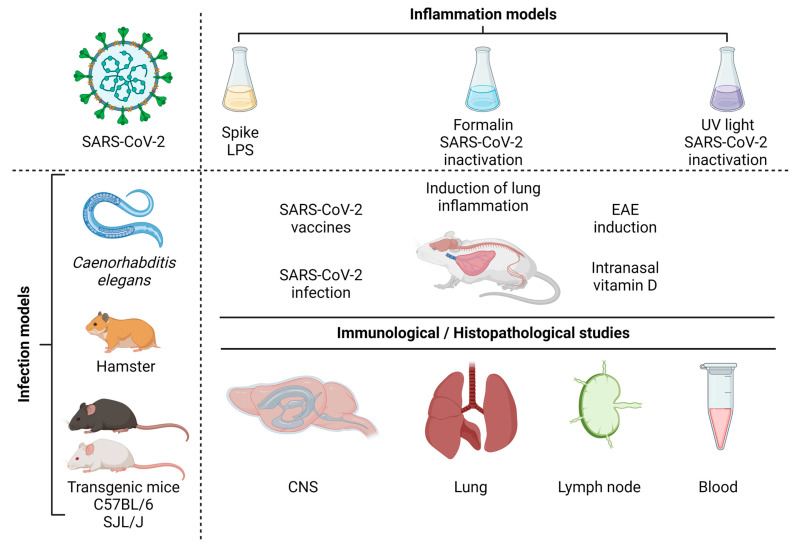
Infection models and immunological/histopathological studies. Source: Created with Biorender.com.

**Table 1 cells-12-00684-t001:** Impact of vitamin D on immune response cells. The arrows ↓ and ↑ indicate reduction or increase in expression, respectively.

Cell Type/Source	Experimental Model	Treatment	Main Outcome
Dendritic cells PENNA and ADORINI 2000 [110] BERER et al., 2000 [111]	Cell culture from peripheral blood monocytes	1,25 (OH)_2_D_3_ added to the culture medium	Inhibition of differentiation and maturation Apoptosis induction ↓ MHC II, CD40, CD80, CD86, IL-12 and IL-23 ↑ IL-10 expression
Macrophages VERWAY et al., 2013 [112] LIU et al., 2006 [113] GOMBARD; BORREGAARD; KOEFLER, 2005 [114]	Co-culture of macrophages and human lung epithelial cells Culture of bone marrow cells from humans and mice	1,25 (OH)_2_D_3_ added to the culture medium	↑ IL-β, IL-8, TNF-α, CCL3, CCL4 and CCL8 ↓ TLR2 and TLR4 Induces cathelicidin synthesis
Peripheral mononuclear cells KHOO et al., 2011 [115]	Human peripheral blood cell culture	1,25 (OH)_2_D_3_ added to the culture medium	↓ Dose-dependent IL-6, TNF-α and IFN-y ↑ cathelicidin
Neutrophils YANG et al., 2015 [116] CHEN; EAPEN; ZOSKI 2015 [117] ARAZ-CIBRIAN; GIRALDO; URCUQUI-ICHIMA 2019 [118]	Human peripheral blood cell culture	1,25 (OH)_2_D_3_ added to the culture medium	↑ Apoptosis in chronic obstructive pulmonary disease ↑ IL-8 levels ↑ NETs formation
Eosinophils MATHEU et al., 2003 [119]	Knockout mice	Vitamin D subcutaneous injection	Eosinophilic narrowing of the upper airways ↓ IL-5 synthesis
Mast cells BIGGS et al., 2010 [120]	Knockout	Vitamin D subcutaneous injection	Eosinophilic narrowing of the upper airways ↓ IL-5 synthesis
Th1 cells SKROBOT; DEMKOW; WACHOWSKA 2018 [121] RAUSCH-FAN et al., 2002 [122]	Human peripheral blood cell culture	1,25 (OH)_2_D_3_ added to the culture medium	↓ Synthesis IL-2, IFN-y, TNF-α Inhibition of IL-12 synthesis
Th2 cell SKROBOT; DEMKOW; WACHOWSKA 2018 [121] BOONSTRA et al., 2001 [123]	Knockout mice		↑ Synthesis of IL-4, IL-5, IL-10 ↑ Transcription of GATA3
Th17 cells IKEDA et al. 2010 [124] JOSHI et al., 2011 [125]	Human peripheral blood cell culture Knockout mice	IP treatment with 1,25 (OH)_2_D_3_ 1,25 (OH)_2_D_3_ added to the culture medium	↓ Synthesis of IL-17, IL-21 and IL-22
Promotion of regulatory T cell differentiation URRY et al., 2012 [126] KANG et al., 2012 [127]	Human peripheral blood cell culture Knockout mice (tissue culture)	1,25 (OH)_2_D_3_ added to the culture medium	↑ Synthesis of IL-10 and of FoxP3 transcription factor
B cells CHEN et al., 2007 [128]	Human peripheral blood cell culture	1,25 (OH)_2_D_3_ added to the culture medium	↓ B cell maturation into plasmocytes and memory cells ↓ Isotype switch
Multiple sclerosis SLOKA et al., 2011 [129] CHAG et al., 2010 [130] COSTA et al., 2016 [131]	Human peripheral blood cell culture Knockout mice (tissue culture)	1,25 (OH)_2_D_3_ administered IP in mouse 1,25 (OH)_2_D_3_ added to the culture medium	↑Th2 ↓ Th1, Th17, IFN-y and IL-17
Rheumatoid arthritis ZHOU et al., 2019 [132]	Knockout mice	IP treatment with 1,25 (OH)_2_D_3_ 1,25 (OH)_2_ D_3_ administered together with the chow	Stopped disease progression ↓ IL-17 and ↑ Tregs
Systemic Lupus Erythematosus ABOU-RAYA; ABOU-RAYA; HELMII 2013 [133] PIANTONI et al., 2015 [134]	Measurement of serum calciferol levels in humans Human peripheral blood cell culture	Oral supplementation with cholecalciferol	↓ IL-8, IL-1, IL-6 and TNF-α ↑ Tregs
Inflammatory bowel disease DANIEL et al., 2008 [135] BARTELS et al., 2007 [136] CANTORNA et al., 2000 [137]	Human peripheral blood cell culture BALB/c mice	1,25 (OH)_2_D_3_ added to the culture medium IP treatment with 1,25 (OH)_2_D_3_	↑ IL-10, IL-4, TGF-β ↓ Th1, IFN-y and TNF-α
Airway Diseases PFEFFER and HAWRYLOWICZ 2018 [138] BREHM et al., 2010 [139] GUPTA et al., 2014 [140] URRY et al., 2012 [126] SUBRAMANINA; BERGMAN; NORMAK 2017 [141]	Serum vitD dosage Asthmatic children Knockout mice	1,25 (OH)_2_D_3_ added to peripheral blood culture and to co-culture of neutrophils and pneumococcus	Low vitamin D levels associated with severe asthma ↑ Tregs and IL-10 ↓ IgE

## Data Availability

Not applicable.

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
