# Peer review of "COVID-19 and Multiple Sclerosis: A Complex Relationship Possibly Aggravated by Low Vitamin D Levels"

_cells, 2023, doi:10.3390/cells12050684_

Round 1

Reviewer 1 Report

The Manuscript is very well written, useful and presents a new idea. The writing does not intend to give a therapeutic strategy, but rather points to consider to unite MS and COVID, revealing the possibility of additional therapeutic strategies.

I believe that the text is developed giving a lot of emphasis to vitamin D, which is done in a very appropriate way. However, when one reads the title this is not clear. The title proposes a general approach, with a broad approach to its pathophysiology and possible therapy. But when one is reading the abstract, one realizes that the focus of the manuscript has a lot to do with vitamin D. When reading the pathophysiology part, the development is wide, but as one advances in the text, the manuscript begins to turn around vitamin D. I reiterate that this development is appropriate and seems good to me. But I think that since vitamin D is an important part of the manuscript, this should be mentioned and made clear in the title. With this, the reader will know that much of what is developed is based on vitamin D and it will not be a surprise that the reader will discover as the reading progresses. Therefore, I suggest somehow adding vitamin D in the title.

In the several sections where he mentions the inflammasome system: “This strong inflammatory process is largely mediated by the inflammasome system.”……” By studying moderate and severe COVID-19 patients, [23], found active NLRP3 inflammasome in PBMCs and tissues of postmortem patients. They also observed correlation of serum inflammasome-derived products, such as Casp1p20 and IL-18 with the markers of COVID-19 severity” In this sense, the manuscript: 1) Novel Coronavirus-Induced NLRP3 Inflammasome Activation: A Potential Drug Target in the Treatment of COVID-19 (doi: 10.3389/fimmu.2020.01021) shows that, unlike other NSAIDs, fenamates (mefenamic , flufenamic acid) selectively inhibits the NLRP3 inflammasome and IL-1β release, and discusses NLRP3 inflammasome inhibitors in the context of inflammatory diseases and COVID; In addition, the manuscript 2) “Efficacy of the use of mefenamic acid combined with standard medical care vs. standard medical care alone for the treatment of COVID‑19: A randomized double‑blind placebo‑controlled trial” (https://doi.org/10.3892/ijmm.2022.5084) states that Mefenamic acid has been shown to selectively inhibit the NLRP3 inflammasome , and that the administration of this anti-inflammatory has a 16‑fold higher probability of achieving acceptable state of health on day 8, compared with the placebo plus standard medical care group. Since they mention the important role of NLRP3 inflammasome, they could refer to the use of an anti-inflammatory, such as mefenamic acid, which acts strongly at that level and helps to make the disease shorter.

Similarly, in the manuscript the presence of long COVID and its neurological manifestations are emphasized in several sections. Likewise, the relevant role of the intestinal microbiota on neurological alterations (related or not to sequelae of COVID) is mentioned. An article was recently published detailing the risk factors for developing long-term COVID. The Guzman-Esquivel et al. (https://doi.org/10.3390/healthcare11020197), points out that one of the important factors for sequelae is the use of antibiotics in the acute phase of COVID. This manuscript postulates limiting the use of antibiotics in acute COVID, or re-establishing the intestinal flora, if it was necessary to give antibiotics, as a proposal to reduce the probability of sequelae. On the same topic, the manuscript by Alper Bilgic et al. (doi: 10.3390/jof8030271) hypothesize that excessive antibiotic administration, which profoundly affects the human microbiota, carries additional health risks that aggravate the health status of COVID patients. For this reason, it would be convenient to highlight in your manuscript not only the role of the microbiota, but perhaps to avoid the alteration of the microbiota with the unjustified use of antibiotics. I believe that the information in these reports could be useful in various parts of your manuscript and should be cited.

Author Response

Cells (ISSN 2073-4409)

Manuscript ID

cells-2168517

Dear reviewer 1,

Thank you very much for the attention given to our manuscript. Your observations and suggestions were accepted because they are clearly relevant. Your constructive considerations allowed us to present a more complete, more clear and hopefully, more useful information to face the challenges of this research area. We deeply appreciate your efforts to improve the quality of this review.

Please, find the answers to all your questions/suggestions below.

  1. (x) English language and style are fine/minor spell check required

The text has been totally revised considering English language and style.

  1. I believe that the text is developed giving a lot of emphasis to vitamin D, which is done in a very appropriate way. However, when one reads the title this is not clear. The title proposes a general approach, with a broad approach to its pathophysiology and possible therapy. But when one is reading the abstract, one realizes that the focus of the manuscript has a lot to do with vitamin D. When reading the pathophysiology part, the development is wide, but as one advances in the text, the manuscript begins to turn around vitamin D. I reiterate that this development is appropriate and seems good to me. But I think that since vitamin D is an important part of the manuscript, this should be mentioned and made clear in the title. With this, the reader will know that much of what is developed is based on vitamin D and it will not be a surprise that the reader will discover as the reading progresses. Therefore, I suggest somehow adding vitamin D in the title.

The title was modified, the new title is:

Covid-19 & Multiple Sclerosis: a complex relationship possibly aggravated by low vitD levels

  1. In the several sections where he mentions the inflammasome system: “This strong inflammatory process is largely mediated by the inflammasome system.”……” By studying moderate and severe COVID-19 patients, [23], found active NLRP3 inflammasome in PBMCs and tissues of postmortem patients. They also observed correlation of serum inflammasome-derived products, such as Casp1p20 and IL-18 with the markers of COVID-19 severity” In this sense, the manuscript: 1) Novel Coronavirus-Induced NLRP3 Inflammasome Activation: A Potential Drug Target in the Treatment of COVID-19 (doi: 10.3389/fimmu.2020.01021) shows that, unlike other NSAIDs, fenamates (mefenamic , flufenamic acid) selectively inhibits the NLRP3 inflammasome and IL-1β release, and discusses NLRP3 inflammasome inhibitors in the context of inflammatory diseases and COVID; In addition, the manuscript 2) “Efficacy of the use of mefenamic acid combined with standard medical care vs. standard medical care alone for the treatment of COVID‑19: A randomized double‑blind placebo‑controlled trial” (https://doi.org/10.3892/ijmm.2022.5084) states that Mefenamic acid has been shown to selectively inhibit the NLRP3 inflammasome , and that the administration of this anti-inflammatory has a 16‑fold higher probability of achieving acceptable state of health on day 8, compared with the placebo plus standard medical care group. Since they mention the important role of NLRP3 inflammasome, they could refer to the use of an anti-inflammatory, such as mefenamic acid, which acts strongly at that level and helps to make the disease shorter.

The following informations were inserted at section 2. COVID-19: clinical manifestations, etiology, and immunopathogenesis:

This crucial inflammasome role in COVID-19 pathogenesis has been investigated as a potential target for therapy. To that end, a plethora of inflammasome inhibitors, including natural products as well as already authorized drugs, should be tested in pre-clinical and clinical studies [22]. Even though future studies are still required, clinical findings otained in a randomized and double-blind placebo-controlled trial in which Mefenamic acid was administered to ambulatory patients [23] significantly reduced their symptomatology in comparison to the placebo group. This efficacy was attributed to both, the anti-viral and the anti-inflammatory properties of Mefenamic acid. (Lines 150-157)

  1. Similarly, in the manuscript the presence of long COVID and its neurological manifestations are emphasized in several sections. Likewise, the relevant role of the intestinal microbiota on neurological alterations (related or not to sequelae of COVID) is mentioned. An article was recently published detailing the risk factors for developing long-term COVID. The Guzman-Esquivel et al. (https://doi.org/10.3390/healthcare11020197), points out that one of the important factors for sequelae is the use of antibiotics in the acute phase of COVID. This manuscript postulates limiting the use of antibiotics in acute COVID, or re-establishing the intestinal flora, if it was necessary to give antibiotics, as a proposal to reduce the probability of sequelae. On the same topic, the manuscript by Alper Bilgic et al. (doi: 10.3390/jof8030271) hypothesize that excessive antibiotic administration, which profoundly affects the human microbiota, carries additional health risks that aggravate the health status of COVID patients. For this reason, it would be convenient to highlight in your manuscript not only the role of the microbiota, but perhaps to avoid the alteration of the microbiota with the unjustified use of antibiotics. I believe that the information in these reports could be useful in various parts of your manuscript and should be cited.

The following informations were inserted at section 2. COVID-19: clinical manifestations, etiology, and immunopathogenesis:

4.1. In addition to the direct virus damage and the deleterious immune response, pieces of evidences reinforce the view that the gut-lung axis will affect both, susceptibility and efficacy of the immune response against the virus. It is well known that the virus affects mainly the respiratory system, however, the gastrointestinal system is also a critical target. Gastrointestinal manifestations such as nausea, vomiting and diarrhea are present in a high percentage of COVID-19 patients. These symptoms have been attributed to the infection of gut epithelial cells by SARS-CoV-2 and the local dysbiosis characterized by alterations in microbiota bacterial composition and diversity [25]. The respiratory tract has its own microbiota and it was already demonstrated that infections by other respiratory virus induce local inflammation which contributes to gut  dysbiosis [26]. A similar effect could be expected from lung SARS-CoV-2 infection. (Lines 166-176)

4.2. Many patients have reported the persistence of symptoms as fatigue, exercise intolerance, dyspnea, muscle pain, insomnia, chest pain, anosmia, cough, and brain fog after the acute disease stage [27]. This condition has been denominated Post-COVID-19 syndrome or Long-COVID-19. Interestingly, in addition to the degree of infection severity, antibiotic usage has been considered one of the main risk factors for Long-COVID-19 development [28]. According to them, antibiotic prescription, which is expected to be more common in severe COVID-19 patients, would alter the gut microbiota composition. This hypothesis is strongly sustained by evidence showing that antibiotics are major disruptors of gut microbiota [29]. In addition, gut dysbiosis triggered by excessive antibiotic administration, together with poorly controlled glycaemia and not well-regulated steroid administration were also pointed as risk factors for COVID-19 associated mucormycosis [30], a rare and lethal fungal infection. Despite the complex gut dysbiosis scenario induced by the virus itself, as demonstrated in both, hamster experimental model and human patients [31,32], which is aggravated by antibiotic use, there is already a variety of promising microbiota-oriented strategies being suggested as prophylactic or therapeutic interventions as probiotics, prebiotics, microbiota-derived metabolites and even fecal transplantation [33]. Besides the microbiota-mediated gut-lung communication axis, another important systemic axis of immune communication impacts COVID-19 and MS: the gut-brain axis, as addressed afterward in this review. Notably, whether lung dysbiosis associated to SARS-CoV-2 infection or antibiotic usage impacts the poor outcomes of COVID-19 is still an open question.   Besides the microbiota-mediated gut-lung communication axis, another important systemic axis of immune communication impacts COVID-19 and MS: the gut-brain axis, as addressed afterward in this review. (Lines 177-199)

The following informations were inserted at section 3. Neurological involvement associated to COVID-19

4.3 As previously addressed in item 2 of this review, dissemination of the virus to the gastrointestinal system is an aggravating condition that can also affect the nervous system due to an altered microbiota gut-brain axis. Of note, gut-microbiota signatures shared by COVID-19 patients and neurological and psychiatric disorders have been described [48]. Such signatures are characterized by a reduction in microbial diversity and richness, expansion of opportunistic proinflammatory pathogens and reduction in anti-inflammatory promoting bacteria. One of the consequences of this disrupted axis is decreased secretion of short-chain fatty acid (SCFA), whose anti-inflammatory ability is well recognized. Therefore, the potential benefit of direct SCFA supplementation or relying on probiotics prescription is being suggested for COVID-19 patients [49]. The disturbed synthesis of other gut-brain axis mediators as, for example, cytokines, 5-hydroxytryptamine, and cholecystokinin can additionally contribute to neurological manifestations during COVID-19 [50]. Notably, the gut-brain axis is also dysfunctional in MS [51], disclosing another link in this already puzzling interplay. (Lines 242-255)

 Author Report Rating(Optional)

We considered the reviewer´s work as very helpful to improve the review´s  content and  clarity;  we rated his work as Excelent.

Reviewer 2 Report

The article entitled "Covid-19 & Multiple Sclerosis relationship: a two-way road" is an extensive review of literature which is associating the COVID-19 with Multiple Sclerosis. Moreover, this article is also talking the possible roles of Vitamin D and its therapeutic potentials. 

However, I have the concerns and suggestions to be addressed before the publication

My first concern is the abstract:

Abstract is actually mixed up and difficult to follow for example:

1. In this sense, will be highlighted their association with inflammation, autoimmunity and central nervous system dysfunction.

inflammation in COVID-19? will be associated with autoimmune disorders

inflammation of which disease will be associated?

2. The possible link of both pathologies with higher risk of development and severity with low vitamin D levels and the possible inclusion of vitamin D supplementation as an adjunct therapy will also be addressed.

Are you going to talk about the roles of therapeutic potentials of Vitamin D in COVID-19?

or over all in both the diseases

or in MS, and CNS dysfunctions.

Please try to fragment it and make the abstract more clear. 

Introduction

Line 21: COVID-19 is a pandemic infection

Check is it correct to write pandemic infection or just infection or disease. 

Line 27: COVID19 to COVID-19, and keep it similar throughout the manuscript. 

In the section no. 2: The authors have described the infection mechanism, and explained about the cytokine storm. No doubt, range of articles have discussed on this aspect previously but the authors have touched some interesting points. However, the Figure no. 2 is very generic, 

It is just showing the entry of the virus into the host cell through ACE2 receptors, I suggest to show that how immune response works after the entrance into the host cell and leads to the cytokine storm.  

The following article can be used.

https://doi.org/10.3390/vaccines11010101

Personally, I find that the section 4 Connection between Covid-19 and autoimmunity is not at all relevant here

Please try to stick with the one theme, 

and try to structure the article in a easily follow up fashion. 

Move this Section 5. Connection between COVID-19 and vitamin D before the section 8. 

So, it will look like the authors first explained the association between MS and COVID-19.

And then focussing on the role of Vitamin D with COVID-19 and MS.

In this way the information will not look like scattered. 

In the section 5:

I suggest to incorporate the following information 

The adequate levels of vitamin D in the host have been associated with the reduced release of proinflammatory cytokines, thus lowering the risk of a cytokine storm; increased levels of anti-inflammatory cytokines and enhanced secretion of natural antimicrobial peptides. It may also be involved in the enhancement of the Th2 immune response and activation of defensive cells such as macrophages. Contrary to these findings, several studies have concluded that there is no direct association between vitamin D concentrations and poor prognosis of the disease [reference: https://doi.org/10.1080/21645515.2022.2025734]. 

And cite the Figure no. 5, as you are showing the up-regulation of the Th2 immune response, T regulatory cells and down regulation of Th1 type cells. 

This will make the article more readable. 

At the end I suggest to improve the conclusion holistically, and try to conclude the information separately, while avoiding the mixing. And try to skip the autoimmunity from the article.

As there is enough to associate with each other. Focus on COVID-19, its association with MS, Then association of these disease with Vitamin D and then the experimental models to study these parameters efficiently in future.

Make most of these revisions round to improve the article sufficiently. 

Best Wishes. 

Author Response

Cells (ISSN 2073-4409)

Manuscript ID

cells-2168517

Dear reviewer 2,

Thank you very much for the attention given to our manuscript. Your observations and suggestions were accepted because they are clearly relevant. Your constructive considerations allowed us to present a more complete, more clear and hopefully, more useful information to face the challenges of this research area. We deeply appreciate your efforts to improve the quality of this review.

Please, find the answers to all your questions/suggestions below.

1.(x) Moderate English changes required

The text has been totally revised considering English language and style.             

2.The article entitled "Covid-19 & Multiple Sclerosis relationship: a two-way road" is an extensive review of literature which is associating the COVID-19 with Multiple Sclerosis. Moreover, this article is also talking the possible roles of Vitamin D and its therapeutic potentials. However, I have the concerns and suggestions to be addressed before the publication. My first concern is the abstract. Abstract is actually mixed up and difficult to follow for example: In this sense, will be highlighted their association with inflammation, autoimmunity and central nervous system dysfunction; inflammation in COVID-19? will be associated with autoimmune disorders, inflammation of which disease will be associated? The possible link of both pathologies with higher risk of development and severity with low vitamin D levels and the possible inclusion of vitamin D supplementation as an adjunct therapy will also be addressed. Are you going to talk about the roles of therapeutic potentials of Vitamin D in COVID-19? or over all in both the diseases or in MS, and CNS dysfunctions. Please try to fragment it and make the abstract more clear.

The abstract was totally rewritten and the new version is transcribed below:

Abstract: Severe acute respiratory syndrome coronavirus 2 (SARS-CoV-2) is an exceptionally transmissible and pathogenic coronavirus that appeared in the end of 2019 and triggered a pan-demic of acute respiratory disease, known as coronavirus disease 2019 (COVID-19). COVID-19 can evolve into a severe disease associated with immediate and delayed sequelae in different organs, including the central nervous system (CNS). A topic that deserves attention in this con-text is the complex relationship between SARS-CoV-2 infection and multiple sclerosis (MS). Here, we initially described the clinical and immunopathogenic characteristics of these two illnesses, accentuating the fact that COVID-19 can, in defined patients, reach the CNS, the target tissue of the MS autoimmune process. The well-known contribution of viral agents such as the Epstein-Barr virus and the postulated participation of SARS-CoV-2 as a risk factor for MS triggering or worsening are then described. We emphasize the contribution of vitamin D in this scenario, considering its relevance in susceptibility, severity and control of both pathologies. Finally, we discuss the experimental animal models that could be explored to better understand the complex interplay of these two diseases, including the possible use of vitamin D as an adjunct immunomodulator to treat them. (Lines 8-21)

keywords: SARS-CoV-2, COVID-19, Multiple Sclerosis, immunopathogenesis, vitamin D

  1. Introduction

Line 21: COVID-19 is a pandemic infection. Check is it correct to write pandemic infection or just infection or disease.  It seems that “pandemic infection” is not used. Infection or disease were adopted throughout the text.

Line 27: COVID19 to COVID-19, and keep it similar throughout the manuscript. COVID-19 was adopted

  1. In the section no. 2: The authors have described the infection mechanism, and explained about the cytokine storm. No doubt, range of articles have discussed on this aspect previously but the authors have touched some interesting points. However, the Figure no. 2 is very generic, it is just showing the entry of the virus into the host cell through ACE2 receptors, I suggest to show that how immune response works after the entrance into the host cell and leads to the cytokine storm.

The following article can beused.https://doi.org/10.3390/vaccines11010101

Original figures 1 and 2 were merged into a single figure 1 (A and B) and a new figure was created to illustrate the immunological changes (Fugure 2), as suggested.

5.Personally, I find that the section 4 Connection between Covid-19 and autoimmunity is not at all relevant here. Please try to stick with the one theme, and try to structure the article in a easily follow up fashion. Move this Section 5. Connection between COVID-19 and vitamin D before the section 8. So, it will look like the authors first explained the association between MS and COVID-19. And then focussing on the role of Vitamin D with COVID-19 and MS.In this way the information will not look like scattered.

The original ideia of including a topic concerning COVID-19 and autoimmunity was to contextualize  multiple sclerosis as an autoimmune disease in the range of autoimmune diseases possibly associated with COVID. But we agree with you that it would extrapolate the main objective of the review and that it would disperse the reader. Therefore, this section was deleted and the other sections were rearranged.  The ensuing sequence was adopted:

1.Introduction 

  1. COVID-19: clinical manifestations, etiology, and immunopathogenesis
  2. Neurological involvement associated to COVID-19
  3. Multiple Sclerosis: clinical manifestations and immunopathogenesis
  4. Connection between MS and virus (EBV and SARS-CoV-2)
  5. Connection between multiple sclerosis and vitamin D
  6. Connection between COVID-19 and vitamin D
  7. Experimental animal models to decipher the complex COVID-19 & MS interplay
  8. Conclusions

  1. In the section 5: I suggest to incorporate the following information:

The adequate levels of vitamin D in the host have been associated with the reduced release of proinflammatory cytokines, thus lowering the risk of a cytokine storm; increased levels of anti-inflammatory cytokines and enhanced secretion of natural antimicrobial peptides. It may also be involved in the enhancement of the Th2 immune response and activation of defensive cells such as macrophages. Contrary to these findings, several studies have concluded that there is no direct association between vitamin D concentrations and poor prognosis of the disease [reference: https://doi.org/10.1080/21645515.2022.2025734].

The following  text was included:

The adequate levels of vitD in the host have been associated with the reduced release of proinflammatory cytokines, thus lowering the risk of a cytokine storm; increased levels of anti-inflammatory cytokines and enhanced secretion of natural antimicrobial peptides. It may also be involved in the enhancement of the Th2 immune response and activation of defensive cells such as macrophages, as illustrated in Figure 5. Contrary to these findings, several studies have concluded that there is no direct association between vitD concentrations and poor prognosis of the disease [161]. (Lines 551-558)

  1. At the end I suggest to improve the conclusion holistically, and try to conclude the information separately, while avoiding the mixing.

The conclusions were rewritten and the new version is transcribed below:

COVID-19 and MS are associated with several immunological disturbances that could, theoretically, interfere with each other disease onset or outcome. SARS-CoV-2 displays, for example, molecular mimicry with CNS epitopes and causes microbiota and BBB disruption which are crucial for MS development. This virus can also reach and inflame the CNS itself which is the target of the autoimmune inflammatory reaction that characterizes MS. These two pathologies also share a possible type I IFN deficient production and hyperactivation of both, the Th1/Th17 axis and the NLRP3 inflammasome platform which could mutually cause disease aggravation. The role of vitD levels in susceptibility, severity, and possible adjunctive therapy in both diseases are investigated and highly discussed but not well-established yet. This complex interplay between COVID-19 and MS urgently needs further and in-depth investigations. A plethora of experimental animal models, usually employed to study each of these pathologies individually, as is the case of C. elegans, hamster strains and transgenic mice, could be explored to investigate aspects related to both diseases simultaneously. These disease models could not only complement the current knowledge but also the possible future questions, bearing in mind that more severe neurological changes associated with long-term COVID are possible. (Lines 718-733)

  1. As there is enough to associate with each other. Focus on COVID-19, its association with MS, Then association of these disease with Vitamin D and then the experimental models to study these parameters efficiently in future.

We believe that the new sequence and the adopted modifications meet these requirements.

 Author Report Rating(Optional)

We considered the reviewer´s work as very helpful to improve the review´s  content and  clarity; we rate this review report as Excelent.

Round 2

Reviewer 2 Report

I appreciate the efforts given to improve the manuscript. The article has been sufficiently revised and can be accepted for the publication.

Best Wishes